# Assessment of Natural Waxes as Stabilizers in Peanut Butter

**DOI:** 10.3390/foods11193127

**Published:** 2022-10-08

**Authors:** Md. Jannatul Ferdaus, Rycal J. S. Blount, Roberta Claro da Silva

**Affiliations:** Department of Family and Consumer Sciences, North Carolina A&T State University, Greensboro, NC 27411, USA

**Keywords:** peanut butter, rice bran wax, carnauba wax, stabilizer, storage quality, texture

## Abstract

Manufacturers add sugar and fully hydrogenated vegetable oils to peanut butter to avoid its oil separation during storage. Unfortunately, hydrogenated oils are significant sources of saturated fats, and reducing their consumption is challenging for food scientists without affecting the desired characteristics of food products. Therefore, in a preliminary study, 1%, 1.5%, and 2% of three natural waxes (rice bran, carnauba, and beeswax) were added to the natural peanut butter to test their efficacy as a stabilizer. Rice bran and carnauba wax added to peanut butter presented a higher elastic modulus (G’) and lower oil separation percentages than beeswax. However, no significant differences were found between the different percentages of waxes. Thus, in the final experiments, 1% of these selected waxes (rice bran and carnauba waxes) were added directly to the roasted ground peanut. Due to the difficulty of adding high melting point waxes to the peanut butter, a second experiment added wax oleogel (rice-bran and carnauba wax) to defatted peanut flour. After four weeks of storage, all of the samples were examined for their texture (TPA) and oil separation. The sample with directly added bran wax had the highest values for spreadability and firmness, and the lowest oil separation, which was 11.94 ± 0.90 N·s^−1^, 19.60 ± 0.71 N·s^−1^, and 0.87 ± 0.05%, respectively. In the peanut flour sample, the spreadability, firmness, and separated oil of the rice bran wax oleogel added sample were 46.95 ± 0.99 N·s^−1^, 66.61 ± 0.93 N, and 1.57 ± 0.07%, respectively. However, the textural properties of the rice bran wax oleogel added sample were close to the commercial peanut butter (natural and creamy). Therefore, the results indicate that the rice bran wax oleogel could be the potential replacement of the fully hydrogenated oil as a stabilizer.

## 1. Introduction

Peanuts have a nutrient-dense profile and are low in simple carbs, despite their high-calorie content. Therefore, they are an excellent source of healthy fats (46.67%), and good source of proteins (26.67%), carbohydrates (16.67%), and fibers (10%) [1]. Furthermore, they are legumes and are considered oilseeds due to their high oil content. In addition, they provide more protein than any other nut, being equivalent to or superior to beans [2]. They are good sources of essential minerals, such as calcium, copper, phosphorus, and zinc, and are also a rich source of the vitamin B complex. Because of their high nutritional value, peanuts are consumed globally in different forms to combat malnutrition. For example, peanut butter-based therapeutic food has been used in the diet program among children in African nations [3]. 

Peanut butter is a paste or spread produced from roasted and ground peanuts, widely used in households and the food industries. Although it is a semi-perishable food product, its storage quality and shelf-life depend on the type of peanut used, added ingredients, manufacturing methods, and storage technique [4]. However, oil separation occurs during peanut butter storage, resulting in natural peanut oil rising to the top and a hard layer of solid peanut remaining at the bottom, making the peanut butter less spreadable. Moreover, this separated oil quickly gets rancid and develops an unpleasant odor [5]. Therefore, in peanut butter production, additives such as salt, sugar, and vegetable oil (stabilizer) are blended with ground peanuts to give the desired quality to the finished product and make it homogenous [6]. Commercially, fully hydrogenated vegetable oils, such as soybean, cottonseed, palm, and rapeseed, are added to peanut butter to prevent oil separation and sensory degradation [7]. These stabilizers develop a matrix that resists oil separation and makes a homogenous peanut butter mixture [8]. Sometimes, these stabilizers induce repulsive force and produce high toughness and low spreadability [9]. Naturally, peanuts are high in unsaturated fatty acids; adding these hydrogenated oils results in an increment of saturated fats in the final product [10]. Saturated fats are well known for raising the amount of LDL (low-density lipoprotein) while lowering HDL (high-density lipoprotein) in the blood. As a result, LDL, or “bad” cholesterol, is deposited on the inner side of arteries, increasing the risk of cardiovascular disease, obesity, and type 2 diabetes [11].

Recently, food-grade waxes have been used in many foods due to their different functional roles. Waxes have stabilization properties and might be used to make peanut butter instead of adding hydrogenated oils [12]. Some previous studies on the waxes have already reported the effectiveness of natural waxes as a stabilizer of peanut butter. Huang et al. applied and reported the capability of rice bran wax as a stabilizer of peanut butter [13]. An experiment by Winkler-Moser et al. used natural waxes such as beeswax (BW), candelilla wax (CLW), rice bran wax (RBW), and sunflower wax (SFW) [14]. This experiment used waxes as an organogelator of oleogel and concluded that they could be potentially used as a stabilizer of peanut butter.

Waxes are animal or plant-derived mixtures of long-chain alcohols and fatty acids with high-melting temperatures [15]. They have a low likelihood of hydrolysis and are poorly digested in the human gastrointestinal tract [16]. Furthermore, waxes do not add saturated fat to the finished peanut butter. Generally, waxes are solid at room temperature and must be melted before being mixed with processed peanuts. When the waxes cool, they create a three-dimensional network that traps and immobilizes the liquid oil, resulting in a semi-solid substance [13]. 

In this experiment, beeswax, rice bran, and carnauba waxes were selected as peanut butter stabilizers. Beeswax is natural, secreted mainly by honeybees to build up their honeycomb. When the wax is secreted from the glands, it is white and then turns a yellowish color, which becomes brown after a few years because of the cocoon [17]. Though the compositions of beeswax depend on the species of bees, their place of wax production, and the age of wax, the main compounds are esters, alkanes, and free fatty acids [18]. In general, beeswax contains 71% of ester compounds, 12–16% of hydrocarbon (C27–C33), 8% of free fatty acids (C24–C32), and 6% of other compounds with melting points ranging from 62 to 65 °C [19,20]. Rice bran wax is naturally derived and obtained during the production process of rice bran oil. In general, rice bran oil contains almost 5% wax, which is extracted to mark the rice bran oil as cooking oil [21]. The major components of rice bran wax are esters with chain length C22–C34, hydrocarbons (C24–C34), free fatty acids (C22–C24), and fatty alcohols (C24–C40). The melting point, acid value, iodine value, and saponification value of rice bran wax are 77–79 °C, 3–8 mg KOH/g, 8–15, and 80–90 mg KOH/g, respectively [22,23]. Carnauba wax is plant-derived and extracted from native Brazilian palm trees. It is made up of many different compounds, including esters with chain length C44–C62, free alcohols with predominant chain length C28–C34, aliphatic and aromatic acids, hydrocarbons, free acids, and diol triterpenes [24]. Ester compounds account for more than 80% of the total, with a predominance of cinnamic acid and aliphatic esters. Moreover, this wax contains a smaller number of secondary alcohols. Its melting temperature is around 83 °C, with saponification and acidity index of 78–90, and 8–10.5 mg KOH/g, respectively [25]. 

The stabilizing process of hydrogenated vegetable oils is analogous to this oleogel system. Moreover, oleogelation is a new concept of structuring edible oil using one or more mixtures of waxes as gelators. In this process, gelators have low solubility in the oil phase, creating a 3D microstructure that traps oil particles [26]. Hence the gelation of vegetable oil occurs, and oleogel is formed, which is solid or semi-solid at room temperature [27]. The oleogels are used to lower the saturated fats, specifically the trans fatty acids in the food system. Some previous studies on oleogel reported the efficiency of natural waxes as crystalline oleogelators because they can crystallize in a well-formed structure with high oil binding capacity (OBC) even at low concentrations [26]. Furthermore, the gelling behaviors (texture and OBC) of the oleogel are determined by the crystal morphology of the used waxes, which is examined in polarized light microscopy (PLM). The crystal morphology depends on the functional group and the chain length of the waxes [28]. Despite the natural waxes’ efficacy as alternative stabilizers in peanut butter, their other qualities have not been well researched. Thus, this research aims to study the effects on the storage quality and textural properties of adding natural waxes (beeswax, rice bran, and carnauba waxes) as a peanut butter stabilizer. 

## 2. Materials and Methods

### 2.1. Materials

Roasted, ground peanuts and peanut flour were purchased from the local market. Rice bran wax (RBW) from Koster Keunen Inc, Watertown, CT, USA, beeswax (BW), and carnauba wax (CNW) from Spectrum Chemical Mfg. Corp., New Brunswick, NJ, USA were selected as stabilizers. Finally, two commercial peanut butters (natural and creamy) were purchased from the local market to compare the texture and oil separation of the samples.

### 2.2. Thermal Properties of Waxes

The melting point of waxes was determined by differential scanning calorimeter (DSC) using the DSC Q20 (TA Instruments, New Castle, DE, USA). 12–15 mg of samples were weighed in a DSC aluminum pan and sealed with a lid. In the DSC, samples were heated at a rate of 5 °C/min from 25 to 80 °C. Samples were kept for 30 min at this temperature, then cooled to −20 °C at a rate of 5 °C/min and kept at this temperature for 90 min. The samples’ melting point (Tm) was determined by the peak temperature of the final melting peak [29].

### 2.3. Preliminary Test Peanut Butter Preparation

Three peanut butter samples were prepared from roasted peanuts following a modified procedure specified by Gills & Resurreccion [4]. Three waxes were added to the peanut butter (BW, RBW, and CNW) in three different ratios (1, 1.5, and 2%). Peanut butter without a stabilizer was prepared as a control sample. The laboratory-prepared samples were kept for four weeks to observe the oil separation percentage. 

### 2.4. Preparation of Peanut Butter from Roasted Ground Peanut

Given the results of preliminary tests, the RBW and CNW (1%) were selected to be studied in the final test. The roasted peanut beans were ground by using Vitamix E310 Blender Explorian Series (Vitamix^®^, Olmsted Township, OH, USA). Waxes were melted at a temperature of 82-85 °C. The melted RBW and CNW waxes (1%) were mixed with warmed peanut butter (R1 and R2) (50–55 °C) separately. Grounded peanuts and waxes were mixed up using a homogenizer 850 (fisher brand^TM^, Waltham, MA, USA) at 10,000 rpm for 2 min. A control sample (R3) was prepared with only the ground peanut. Figure 1 (method 1) represents the process of peanut butter production from roasted ground peanuts.

### 2.5. Preparation of Peanut Butter from Peanut Flour and Oleogels

Due to the difficulties of adding pure waxes to a semi-solid system, such as peanut butter, the second method used defatted peanut flour with wax-oleogel. Peanut flour was bought from a local grocery store and had 13.33% fat. Wax oleogels were prepared using peanut oil as the continuous phase and 3.4% of waxes (RBW and CNW). The peanut oil was used as the continuous phase and the RBW and CNW as oleogelators. The individual waxes (4.25 g) were added to 125 g of heated oil under continuous stirring to make 3.4% oleogel. The peanut oil was heated, and when it reached the melting temperature (82–85 °C) of selected waxes, the oleogelator was added and mixed for 10 min until complete dissolution. Once they (peanut oil and waxes) were melted and mixed together, the heat was removed from the samples, and they cooled to room temperature. Then 300 g of peanut flour was mixed up with the oleogels (F1-RBW oleogel and F2–CNW oleogel) using the homogenizer 850 (fisher brand^TM^, Waltham, MA, USA) for 2 min at 10,000 rpm. The amount of wax in the final peanut butter sample was 1%. Moreover, the fat percentage in each peanut butter was closely maintained, being 42% for the roasted peanuts and 38.82% (including added peanut oil as oleogel) for the peanut flour. The control sample (F3) was prepared from peanut flour (300 g) and peanut oil (125 g). Additionally, the quality of the formulated samples was compared to the commercial natural (C1) and creamy (C2) peanut butter. Figure 1 (method 2) illustrates the preparation of peanut butter from oleogel and peanut flour.

### 2.6. Peanut Butter Properties 

#### 2.6.1. Oil Separation

The oil separation was observed after four weeks of storage, following the method of Radočaj et al. [30]. Pre-weighted filter paper (A) was put on top of each sample (B). Then after 4 weeks of storage, filter papers were collected and weighed (C) to measure the separated oil as a percentage.
$$Oil\;Separation=\left[\frac{C-A}{B}\right]*100$$
where B = weight of the peanut butter.

#### 2.6.2. Viscoelastic Properties 

The viscoelastic properties of the peanut butter samples of the preliminary studies were determined in triplicate using a magnetic bearing rheometer (TA Instruments AR-G2, New Castle, DE, USA) [29]. Experiments were conducted utilizing a concentric cylinder of standard size (15.17 mm diameter-991036) equipped with a temperature-controlled Peltier plate. The stress sweep tests were performed at 25 °C and a constant angular frequency of 1 Hz over a shear stress range of 0.01–100 Pa, which covers both the linear and non-linear viscoelastic range of peanut butter. The dynamic rheological data obtained included the storage modulus (G′). 

#### 2.6.3. Texture Analysis

Spreadability, firmness, and adhesivity of peanut butter and the oleogels of the final experiments were measured according to the method reported by Mohd Rozalli et al. with some modifications [5]. 

A texture analyzer TA. HD PLUS from Stable (Surrey, UK) used a load cell of 5 kg with a conical TTC Spreadability Rig (HDP/SR) attachment (Stable Microsystems, Surrey, UK) consisting of a set of precisely matched male (positive) and female (negative) acrylic 90° cones. The weight and height of the machine were calibrated for 5000 g and 25 mm, respectively. The measurement was carried out at a test speed of 1.0 mm/s and a penetration depth of 25 mm. Each sample was measured five times. Textural analysis of the peanut butter and oleogels were analyzed with a stable micro system texture analyzer in quintuplicate. 

#### 2.6.4. Polarized Light Microscopy

The microstructure of oleogels was investigated by polarized light microscopy (PLM) [31]. A drop from the prepared oleogels was gently placed onto a preheated slide and carefully overlaid with a glass cover slip. Images were acquired using an Olympus BX51 polarizing microscope (Olympus Optical Co Ltd., Tokyo, Japan) equipped with a Linkam thermal system (Linkam Scientific Instruments Ltd., Redhill, UK). At first, the temperature was increased to 100 °C, then decreased to 20 °C by 5 °C/min. Finally, the sample’s temperature was again increased to 100 °C at the same rate.

#### 2.6.5. Oil Binding Capacity

The Oil Binding Capacity (OBC) test is a method to determine the strength of oleogel structure, and how well the oleogel can hold onto oil at a molecular level. Oil loss was evaluated using a centrifugation process where a pre-weighed 2 mL empty Eppendorf (*a*) tube was filled with 1 mL of the sample. The Eppendorf tube (*b*) along with the sample was weighed, then refrigerated for 1 h minimum. The Eppendorf was centrifuged for 15 min at 10,000 rpm after this cooling process. The tubes were then left upside down, allowing unstabilized oil to drain onto a filter paper for 10 min. The drained Eppendorf tube was weighed again (*c*) and calculated according to equation [32].
OBC % Calculation
$$100-\frac{\left[\left(b-a\right)-\left(c-a\right)\right]}{\left(b-a\right)}\;x\;100$$

#### 2.6.6. Statistical Analysis

All measurements were made in triplicate, and primary data were collected in Microsoft Excel 2016. Data from the preliminary test were computed in GraphPad Prism v. 6.0 (GraphPad Prism Software, San Diego, CA, USA) by using one-way ANOVA. However, the data from the final test were analyzed for the significant differences in Statistical Package for the Social Sciences (SPSS) V26 by using an independent sample t-test and one-way ANOVA. In both cases, the mean comparison was set at a confidence level of 95%. 

## 3. Results and Discussion

### 3.1. Preliminary Test

As a result of the oils separation analysis, the natural peanut butter presented 1.8% of oil separation after four weeks under storage (Figure 2a). This is expected behavior in natural peanut butter, and is correlated to the high oleic acid content of peanuts (ca.50%). Oleic-rich oils have a high viscosity that limits the crystal network’s structural formation [33]. 

The waxes thermal profiles (Table 1) showed the BW with a lower melting peak at 65.11 ± 0.18 °C, compared to RBW (83.59 ± 0.35 °C) and CNW (83.02 ± 0.02 °C). The lower temperatures observed in the BW melting curve are associated with its heterogeneous and complex composition, containing 35–45% of monoesters and hydroxy monoesters, 12–16% of hydrocarbons with a predominant chain length of C27–C33, free fatty acids (12%–14%) with a chain length of C24–C32, complex wax esters (15%–27%), and free fatty alcohols (ca. 1%) [34].

The beeswax-added peanut butter showed a high percentage of oil separation after four weeks of storage with 1.09%, 1.3%, and 1.6% of oil separation for 1, 1.5, and 2% of BW addition, respectively (Figure 2a). The same behavior was observed by Winkler-Moser et al., where BW presented a higher oil separation rate compared with other waxes to stabilize peanut butter [12]. The authors reported that the increase of the OBC was significant only with the addition of 2% of BW to the peanut butter, indicating either a lower level of crystallization or that it did not form a crystalline network that was able to effectively bind oil. The lower ability to stabilize the peanut butter could be associated with the heterogeneous nature of BW components with lower order arrangement, which may impede the crystal growth and network formation. 

The addition of RBW and CBW (1, 1.5, and 2%) stabilized the peanut butter through the four weeks of the preliminary study with no oil separation. 

The peanut butter’s viscoelasticity (G’) was observed in the rheology test (Figure 2b). The G’ of natural peanut butter was 278.2 ± 63.9 Pa and the addition of 1% of BW increased the G’ to 1372.9 ± 352.7 Pa, 2226.5 ± 359.8 Pa, and 1780.7 ± 407.2 Pa adding 1.5 and 2% of BW, respectively, with no significant difference (*p* > 0.05). The addition of CNW also increased the viscoelasticity of the peanut butter, with values ranging from 3621.0 ± 1808.0 Pa (1%) to 8867.6 ± 2163.7 Pa (2%). The most prominent results were observed with the addition of RBW with the G’ significantly higher (*p* > 0.05) increased according to the waxes added. The 1% of RBW showed a G’ of 19,773.3 ± 8510.8 Pa and 18,841.2 ± 4410.4 Pa, 20,882.8 ± 5690.3 for 1.5% and 2%, respectively.

However, no significant values were found in their rheology and oil separation test for the different percentages of RBW and CNW. Therefore, in the final experiment, a minimum value (1%) was established in this context.

### 3.2. Oleogels

The textural properties and oil binding capacity are shown in Table 2. The firmness (2.01 ± 0.33 N), toughness (4.12 ± 0.96 J·m^−3^), and OBC% (99.65 ± 0.40) of the 3.4% (*w*/*w*) RBW oleogel were measured to be significantly (*p* < 0.001) higher than the 3.4% (*w*/*w*) CNW oleogel. On the other hand, the adhesiveness of the rice bran wax oleogel was lower (*p* < 0.001) than the CNW oleogel, which were -0.27 ± 0.04 N·s^−1^, and −0.02 ± 0.01 N·s^−1^, respectively. These results showed the contrary relation between the spreadability and adhesiveness of the oleogel, which was previously reported by Martins et al. [35]. Furthermore, in a previous review on oleogel, Co & Marangoni reported in 2012 that oleogel with lower OBC has a higher level of softness [36]. However, the physical characteristics of oleogels, such as oil binding capability (OBC), spreadability, firmness, toughness, and adhesiveness, are influenced by the crystal and microstructure of added waxes [28].

The crystal structures of the oleogel samples were examined at different temperatures in Polarized Light Microscopy (PLM), and the pictures are shown in Figure 3 (RBW-oleogel) and Figure 4 (CNW-oleogel). Waxes are organic substances that often contain free fatty acids, primary and secondary alcohols, aldehydes, and fatty acid esters, among other functional groups [37]. The physical features of the waxes are determined by the combination of these components and some other unique configurations, such as hydrocarbon chain length and the number and position of unsaturated bonds. Although 1% *w*/*w* RBW and peanut oil showed a needle-like structure, CNW oleogel had spherulitic structures, which were also reported by Dassanayake et al. [23]. However, a needle-like shape is expected for the gel formation, which entraps a large amount of oil and other food particles in it [26,38]. The chemical compositions of natural waxes (polarity, types and fatty acid length, and melting point) and external influences such as oil polarity, mixing and cooling temperature, and storage time affect the crystallization of oleogels as their interaction with the food products [39]. The main chemical compositions of RBW are esters of fatty acids (carbon number 16 to 32) and fatty alcohols with carbon numbers from 24 to 38 [22]. That indicates a distinctive characteristic of the RBW with a significant portion of the long-chain fatty acids and alcohol esters and shows a unique crystallization behavior. On the other hand, CNW contains esters of fatty acids (80–85%), fatty alcohols (10–15%), esterified fatty diols (ca. 20%), hydroxylated fatty acids (about 6%), hydroxylated or methoxylated cinnamic acid (ca. 10%), hydrocarbons 1–3%, and resins 4–6% [23]. Thus, the CNW can present multiple melting and crystallization profiles in the oleogel system for many functional groups.

### 3.3. Texture Analysis of Peanut Butter

#### 3.3.1. Roasted Peanut

The samples from roasted peanuts were tested for their textural properties such as spreadability, firmness, and adhesiveness, which are illustrated in Figure 5. Directly adding 1% RBW to the peanut butter sample (R1) showed a higher spreadability (11.94 ± 0.90 N·s^−1^) and firmness (19.60 ± 0.71 N) than the other butter samples from ground roasted peanuts. Moreover, the texture values of R1 and R2 were higher than the natural peanut butter (R3), where the spreadability and firmness of natural peanut butter were 3.34 ± 0.12 N·s^−1^ and 7.18 ± 0.29 N, respectively. However, the spreadability and firmness R1 sample were found to be higher than the C1 and lower than the C2 commercial sample. An experiment by Gills & Resurreccion on peanut butter and hydrogenated vegetable oil reported a spreadability of 111.86 N·s^−1^ [4]. This finding was higher than the spreadability of directly added wax samples and commercial peanut butter in the current experiment. However, as Gills & Resurreccion added hydrogenated vegetable oil, this could be related to the increased spreadability [4]. In a hydrogenated vegetable oil-added peanut butter sample, the firmness recorded was 93.31 ± 14.02 N [40], which was higher than the firmness of the current study. On the other hand, in the adhesiveness test, R1(−4.45 ± 0.30 N·s^−1^) and R2 (−4.10 ± 2.17 N·s^−1^) showed similar results, and these values were lower than the raw peanut butter (R3 = −1.51 ± 0.05 N·s^−1^). Overall, the texture properties results were lower than the commercial peanut butter samples. 

#### 3.3.2. Peanut Flour

In oleogel-added peanut butter samples, F2 had the highest spreadability (115.03 ± 2.44 N·s^−1^) and firmness (115.97 ± 2.49 N) than all other samples (Figure 5). These findings were significantly (*p* < 0.05) higher (Table 3) than all other samples, including commercial peanut butter (C1 and C2). However, these findings were higher than previous experiments of Gills & Resurreccion on peanut butter with hydrogenated vegetable oil, where they recorded spreadability and firmness were 111.86 N·s^−1^ and 101.27 ± 9.00 N respectively [4]. Moreover, the spreadability (46.95 ± 0.99 N·s^−1^) and firmness (46.95 ± 0.99 N) of the RBW-oleogel added (F1) sample was compared and found to be significantly (*p* < 0.05) different than other samples (Table 3). These textural properties of the F1 sample were closer to the commercial creamy peanut butter (C2). For example, the spreadability and firmness of the C2 sample were 28.15 ± 2.71 N·s^−1^ and 47.94 ± 3.77 N, respectively. However, a previous study on the stabilization of peanut butter also reported the same result. In that experiment, Winkler-Moser et al. added 1.0–2.0% (*w*/*w*) beeswax (BW), candelilla wax (CLW), rice bran wax (RBW), and sunflower wax (SFW) to the peanut butter [14]. However, the sample stabilized with 1–1.5% RBW showed the closest results to the commercial reference sample. These significant (*p* < 0.05) differences in spreadability and firmness between RBW and CNW oleogel added butter samples could be explained by the characteristics of oleogel samples. The textural properties and oil binding capacity (OBC) of the two oleogel samples have been shown in Table 2. The OBC of RBW-oleogel and CNW-oleogel were recorded at 99.65 ± 0.40% and 19.78 ± 0.77%, respectively. As the OBC was higher in RBW-oleogel, that is why the spreadability was also higher than the CNW-oleogel. However, when the crystal morphology of the two oleogels was examined in PLM, the RBW and CNW oleogel showed needle-like and spherulitic structures, respectively (Figure 3 and Figure 4). For this structural difference, RBW-oleogel entraps more peanut particles than the CNW-oleogel, hence F1 sample had less spreadability than the sample F2. The lowest adhesiveness was found in commercial creamy peanut butter (C2), which was −10.99 ± 0.85 N·s^−1^. F1 and F3 samples showed relatively the same adhesiveness, which was −7.80 ± 2.63 N·s^−1^, and −8.24 ± 2.73 N·s^−1^, respectively. However, F3 (control sample) had the lowest adhesiveness among the laboratory-prepared peanut butter samples. Peanut butter with more oil exhibited a lower adhesion force and higher work of adhesion and stretch than those with less oil [41]. Adhesiveness measures the force required to remove peanut butter from the palate, teeth, and tongue [42]. 

An experiment was previously conducted to assess the effect of temperature on the spreadability of commercial peanut butter [43]. This study found that the spreadability was 13.19 ± 0.44, 9.84 ± 0.45, and 9.27 ± 0.40 N·s^−1^ at 10, 25, and 35 °C, respectively. Though the finding was higher at 10 °C than the butter from roasted peanuts, it was far lower than the F2 sample in the current study. That study showed that natural peanut butter was identical to commercial peanut butter in terms of spreadability and firmness after being stored at 10 °C for 8 weeks. After the eighth week, it begins to resemble commercial peanut butter. Natural peanut butter was shown to be easier to spread and less stiff than commercial peanut butter after 3 and 6 weeks at higher storage temperatures of 25 and 35 °C, respectively [5]. The peanut grind size affects textural properties; increased grind size decreases spreadability, adhesiveness, and firmness [41]. That is why the differences in particle size between roasted ground peanuts and peanut flour also could be considered for their differences in textural characteristics.

### 3.4. Oil Separation of Peanut Butter

#### 3.4.1. Roasted Peanut

Figure 6 illustrates the percentages of separated oil and its trend in every week of storage. However, R2 and R3 samples showed a considerably increased oil separation trend over 4 weeks of storage. After one week, the separated oil of R2 and R3 was 0.36% and 0.37% (Figure 6a), respectively, and after four weeks, total separation was 1.96% and 1.97% (Figure 6b). However, in the R1 sample, the total oil separation was recorded up to 0.87%, which was the lowest among the butter from roasted peanuts. A study by Aryana et al., incorporated palm oil into peanut butter as a stabilizer [44]. They added 0%, 1.5%, 2%, and 2.5% of palm oil (*v/w*%) to the peanut butter and measured the percentage of separated oil after 15 weeks. For the 0%, 1.5%, 2%, and 2.5% of the palm oil, the separated oil was found 4.95%, 4.86%, 3.98%, and 5.51% at 21 °C, respectively. It proves that the oil separation percentage and trends of the current study are lower compared to Aryana et al. [44]. In peanut butter, separated oil easily gets rancid and deteriorates the quality. This is why peanut butter has the lowest percentage of separated oil and has the highest shelf life.

#### 3.4.2. Peanut Flour

In the butter from peanut flour, the RBW oleogel added sample (F1) showed steady oil separation through the storage period, which was 0.38% and 0.40% in the 1st and 4th week, respectively. The lowest value was found for the CNW oleogel (F2), the total separated oil was 0.80% after 4 weeks. However, when the oil separation of F1 (1.57 ± 0.07%) was compared to other samples, R1 and F2 were found to be significantly (*p* < 0.05) different for using RBW oleogel (Table 3). Furthermore, when F2 was compared to the oil separation percentages of the other samples, R2, R3, and F1 were found to be significantly (*p* < 0.05) different for using CRW oleogel. A previous study on natural peanut butter recorded up to 21% of oil separation after 10 weeks, which is much more than the current experiment [5]. Gills & Resurreccion researched the unsterilized and 2.5% palm oil added peanut butter [4]. In that experiment, they recorded a maximum of 15%, and 17% of oil separation in unsterilized and palm oil added peanut butter, respectively. These oil separation rates are higher than the formulated samples of the current study. On the other hand, this study found a high percentage of oil separation for the directly added carnauba wax (R2) and control (R3) samples from roasted ground peanuts, which were 1.96% and 1.97%, respectively. Moreover, the amount of added wax (1%) was lower in the current study, which is considerably cost-effective for peanut butter production at the industrial level. 

## 4. Conclusions

This research is wholly related to finding an alternative stabilizer for peanut butter and the formulation of plant-based oleogels. Texture and oil separation analysis showed the efficiency level of the waxes as a stabilizer. The consumers’ main desirable characteristics of peanut butter are spreadability and the lowest oil separation. This research showed the potential of waxes as a natural stabilizer;overall the oil separation was lower than 2% after four weeks of storage at room temperature. This means that both pure wax and the wax-oleogel stabilized the peanut butter samples. By considering the spreadability and firmness of the peanut butter, RBW-oleogel added to peanut flour showed a close spreadability and firmness to the commercial peanut butter. This oleogel could be a sustainable, USDA-approved alternative to the “fully hydrogenated oil” as a natural stabilizer. Therefore, oleogel could be a breakthrough in peanut butter production. In the future, this experiment will include the consumer acceptability of the peanut butter sample. 

## Figures and Tables

**Figure 1 foods-11-03127-f001:**
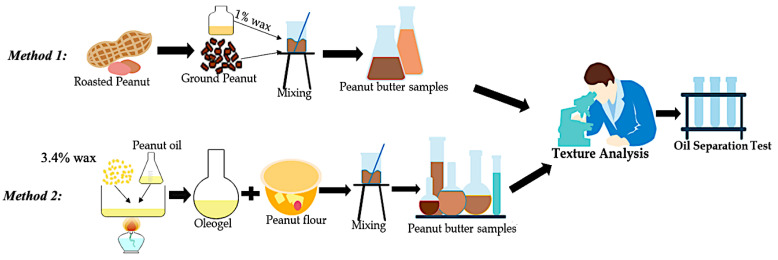
Flow diagram of peanut butter production.

**Figure 2 foods-11-03127-f002:**
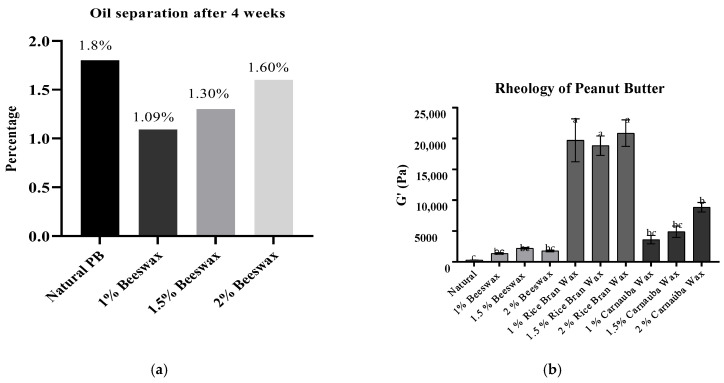
Percentages of used waxes in peanut butter, oil separation, and rheology. (**a**) represents the percentage of added waxes and oil separation of the samples in preliminary test; (**b**) represents the viscoelasticity (G’) of all samples in preliminary test. *a–c*: same letter on the rheology figure means no significant (*p* > 0.05) differences between the values.

**Figure 3 foods-11-03127-f003:**
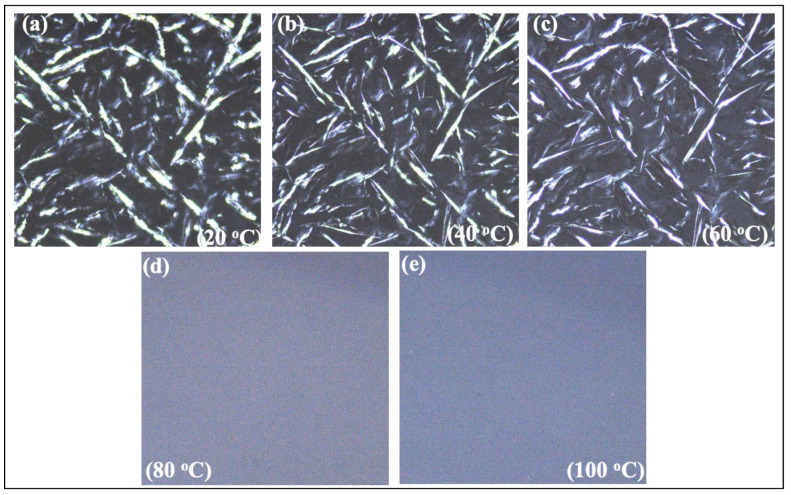
Polarized light microscopy of 3.4% Rice bran wax (RBW) oleogel. (**a**) = wax crystal at 20 °C; (**b**) = wax crystal at 40 °C; (**c**) = wax crystal at 60 °C; (**d**) = wax crystal at 80 °C; (**e**) = wax crystal at 100 °C.

**Figure 4 foods-11-03127-f004:**
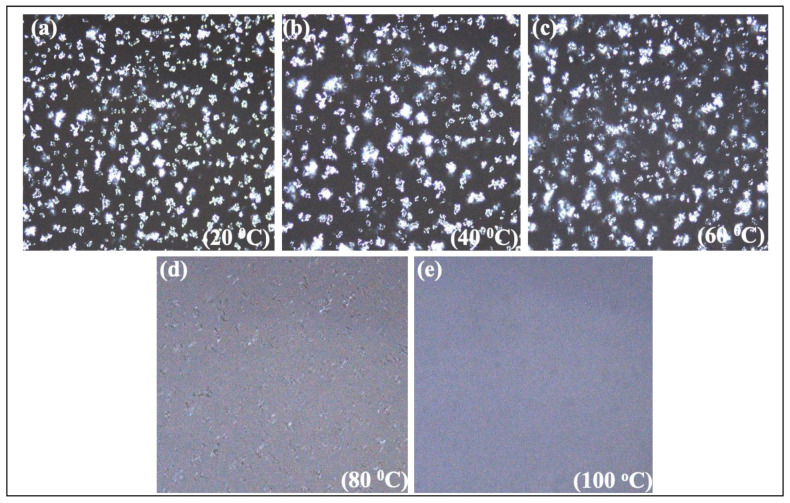
Polarized light microscopy of 3.4% carnauba wax (CNW) oleogel. (**a**) = wax crystal at 20 °C; (**b**) = wax crystal at 40 °C; (**c**) = wax crystal at 60 °C; (**d**) = wax crystal at 80 °C; (**e**) = wax crystal at 100 °C.

**Figure 5 foods-11-03127-f005:**
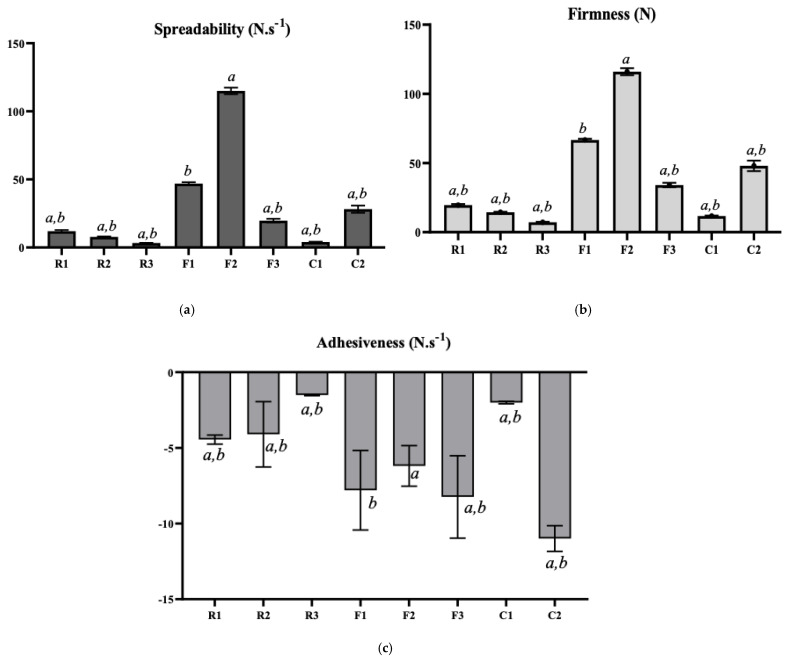
Texture analysis of final peanut butter samples. Values were significantly (*p* < 0.05) different for using rice bran wax oleogel in sample F1; *b =* values were significantly (*p* < 0.05) different for using carnauba wax oleogel in sample F2. R1 = rice bran wax added sample; R2 = carnauba wax added sample; R3 = un-stabilized butter from natural peanut (control); F1 = rice bran wax oleogel added sample; F2 = carnauba wax oleogel added sample; F3 = un-stabilized butter from peanut flour; C1 = commercial natural peanut butter; C2 = commercial creamy peanut butter.

**Figure 6 foods-11-03127-f006:**
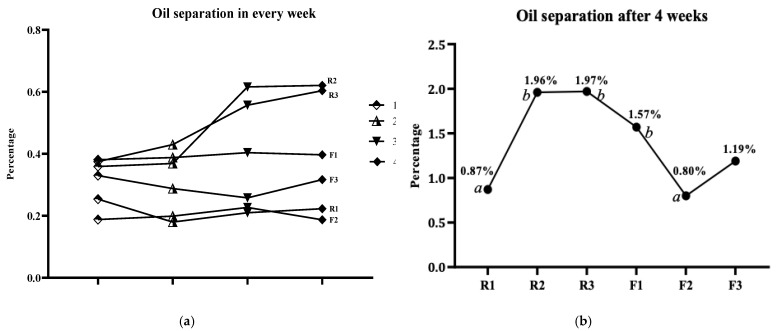
Oil separation percentage of the peanut butter samples. *a =* values were significantly (*p* < 0.05) different for using rice bran wax oleogel in sample F1; *b =* values were significantly (*p* < 0.05) different for using carnauba wax oleogel in sample F2. R1 = rice bran wax added sample; R2 = carnauba wax added sample; R3 = un-stabilized butter from natural peanut (control); F1 = rice bran wax oleogel added sample; F2 = carnauba wax oleogel added sample; F3 = un-stabilized butter from peanut flour.

**Table 1 foods-11-03127-t001:** Melting profile of the three selected waxes.

Name of the Waxes	Onset (Mean ± SD) °C	Peak (Mean ± SD) °C	Delta (Mean ± SD) °C
Rice bran wax	80.74 ± 0.89	83.59 ± 0.35	209.35 ± 3.04
Carnauba wax	80.72 ± 0.21	83.88 ± 0.02	187.75 ± 10.82
Beeswax	58.42 ± 4.21	65.11 ± 0.18	157.88 ± 0.97

**Table 2 foods-11-03127-t002:** Texture and oil binding capacity of oleogels (Mean ± SD).

Parameters	Rice Bran Wax	Carnauba Wax
Spreadability (N·s^−1^) ^***^	1.33 ± 0.32	0.05 ± 0.03
Firmness (N) ^***^	2.01 ± 0.33	0.14 ± 0.03
Toughness (J·m^−3^) ^***^	4.12 ± 0.96	0.17 ± 0.08
Adhesiveness (N·s^−1^) ^***^	−0.27 ± 0.04	−0.02 ± 0.01
OBC% ^***^	99.65 ± 0.40	19.78 ± 0.77

^***^ Values were significantly (*p* < 0.001) different for using rice bran and carnauba waxes in oleogels.

**Table 3 foods-11-03127-t003:** Texture analysis and oil separation (Mean ± SD) of stored peanut butter.

Samples Name	Oil Separation after 4 Weeks (%)	Spreadability (N·s^−1^)	Firmness (N)	Adhesiveness (N·s^−1^)
R1	0.87 ± 0.05 *^a^*	11.94 ± 0.90 *^a,b^*	19.60 ± 0.71 *^a,b^*	−4.45 ± 0.30 *^a,b^*
R2	1.96 ± 0.33 *^b^*	7.79 ± 0.33 *^a,b^*	14.39 ± 0.35 *^a,b^*	−4.11 ± 2.17 *^a,b^*
R3	1.97 ± 0.38 *^b^*	3.34 ± 0.12 *^a,b^*	7.18 ± 0.30 *^a,b^*	−1.51 ± 0.05 *^a,b^*
F1	1.57 ± 0.07 *^b^*	46.95 ± 0.99 *^b^*	66.61 ± 0.93 *^b^*	−7.80 ± 2.63 *^b^*
F2	0.80 ± 0.03 *^a^*	115.04 ± 2.44 *^a^*	115.97 ± 2.49 *^a^*	−6.19 ± 1.34 *^a^*
F3	1.19 ± 0.09	19.70 ± 1.35 *^a,b^*	34.07 ± 1.60 *^a,b^*	−8.24 ± 2.73 *^a,b^*
C1	------	4.01 ± 0.19 *^a,b^*	11.69 ± 0.38 *^a,b^*	−2.01 ± 0.09 *^a,b^*
C2	------	28.15 ± 2.71 *^a,b^*	47.94 ± 3.77 *^a,b^*	−10.99 ± 0.85 *^a,b^*

*a* = values were significantly (*p* < 0.05) different for using rice bran wax oleogel in sample F1; *b* = values were significantly (*p* < 0.05) different for using carnauba wax oleogel in sample F2. R1 = rice bran wax added sample; R2 = carnauba wax added sample; R3 = un-stabilized butter from natural peanut (control); F1 = rice bran wax oleogel added sample; F2 = carnauba wax oleogel added sample; F3 = un-stabilized butter from peanut flour; C1 = commercial natural peanut butter; C2 = commercial creamy peanut butter.

## Data Availability

The data presented in this study are available on request from the corresponding author.

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
