# Peer review of "Assessment of Natural Waxes as Stabilizers in Peanut Butter"

_foods, 2022, doi:10.3390/foods11193127_

Round 1
Reviewer 1 Report
Comments for author:
This manuscript titled “The use of rice bran and carnauba waxes as stabilizers of peanut butter” (ID No. foods-1920811) is meaningful in understanding the waxes as potential stabilizers to improve the stability of peanut butter. The introduction was well written and the purpose of the study was clear. ​However, further improvement is needed to enhance the quality. Some questions and amendments are shown below.
Line 51: "they are an excellent source of healthy fats (20.7-25.3%), proteins (31- 46%)" is wrong and you should correct it.
Figure 1: This figure shows the method of preparation of peanut butter from roasted ground peanut (2.4.1) and the preparation of peanut butter from peanut flour and oleogels (2.4.2). However, you only refer to figure 1 in 2.4.1 and this figure shows the preparation method for 2.4.2 in the first place. This is difficult to understand. I suggest that you label method 1 and method 2 in figure 1 and refer to figure 1 in 2.4.2.
Line 175: How do you remove the fat from the peanut flour? Is it a physical or chemical method? And what is the residual oil rate of peanut flour?
Lines 185-188: How do you calculate to get the fat percentage of peanut butter prepared with peanut flour? I have calculated that the fat percentage in your peanut butter is 29.4% (300g of peanut flour and 125g of peanut oil). Is this right? Meanwhile, what is the basis of your peanut flour to peanut oil ratio?
Line 260: The modulus of loss (G″) is determined in Materials and Methods, but the results for G″ are not discussed, please add.
Figure 2(b): Please add the data of G″ in this figure.
Figure 3: In Fig. 2 the number is 2, but in Fig. 03 the number is 03. The format of the figure should be consistent, and the figures should be reviewed carefully.
Figure: The numbers in the figure are in the wrong order, please correct it.
Lines 396-398: Because the F samples use defatted peanut flour, the oil is sufficiently removed. Whether it will cause a higher degree of oil separation than normally prepared peanut butter?
Lines 419-420: "oleogel could be a breakthrough in peanut butter production." Do you have data on consumer perception of this product?
Table 3: Do the lower oil separation of the C1 and C2 samples compared to the other samples indicate that the other samples are less stable than the commercial natural (C1) and cream (C2) peanut butter?
Author Response
"Please see the attachment"

Reviewer 2 Report
Good work, but the study idea has an average novelty, and several papers were done on the same point (2019- 2022). Therefore, I mentioned several papers on the same point at the end of the comments.
1. Title is very simple and not attractive, I think it will be better if you replace it with "Assessment of natural waxes as stabilizers in peanut butter"
2. the word "strong quality" in keywords is inappropriate so, replace it with "quality"
3. the sequence of the manuscript is very bad and makes me confused, firstly; you should start with waxes properties, then prepare peanut butter as a preliminary or final product to prevent repeating the same method.
4. almost all methods without references
5. there is no figure or table caption to identify used abbreviations, also why you didn't show statistical analysis on the figures?
6. discussion was short and needs to improve
7. line 350: correct the figures number to 5, 6 instead of 05, 06
8. the manuscript is missing the most important evaluation which is the sensory evaluation to determine if it is acceptable for consumers or not.
9. introduction didn't discuss the previous studies that used natural waxes as stabilizers in peanut butter.
previous studies:
1. Stabilization of peanut butter by rice bran wax
Zhaohua Huang 1, Baozhong Guo 1, Chong Deng 1, Shunjing Luo 1, Chengmei Liu 1, Xiuting Hu 1
DOI: 10.1111/1750-3841.15176
2. Evaluation of Beeswax, Candelilla Wax, Rice Bran Wax, and Sunflower Wax as Alternative Stabilizers for Peanut Butter
Jill K. Winkler-Moser,Julie Anderson,Jeffrey A. Byars,Mukti Singh,Hong-Sik Hwang
3. Stabilization of peanut butter by rice bran wax
May 2020 Journal of Food Science 85(3)
DOI: 10.1111/1750-3841.15176
Zhaohua HuangBaozhong GuoBaozhong GuoChong DengShow all 6 authors
4. Texture and flavor evaluation of peanut butter stabilized with natural waxes
Jill K. Winkler-Moser,Julie A. Anderson,Hong-Sik Hwang
First published: 23 March 2022 https://doi.org/10.1111/1750-3841.16118
Author Response
"Please see the attachment"

Round 2
Reviewer 1 Report
Comments for author:
The authors have addressed the questions quite well. The revised manuscript has been improved significantly. There are no further comments.
Author Response
We are grateful to the reviewer for the manuscript's attention and for her/his comments and suggestions.
Reviewer 2 Report
Nice efforts for manuscript modifications, but still one thing I hope you can do it, it's sensory evaluation because I think there is no need for further investigation. All I asked to make small scale sensory evaluation to the samples with the best results after prepration
Author Response
Dear reviewer,
We are grateful for your attention to the manuscript and for your comments and suggestions.
The authors understand that sensory evaluation in the food industry plays a crucial role in new product development and also is essential to assess consumer acceptance of the substitution of ingredients for specific formulations. However, the sensory analysis of peanut butter added of different waxes and oleogels should be studied in a more complete sensory study to address not only the acceptability of consumers but also to understand what drives the consumer preferences for different kinds of peanut butter using descriptive methods. Only with these responses will the authors be able to evaluate and discuss the results of the peanut butter added of waxes and oleogels.
The authors have already submitted an IRB (Institutional Review Boards) proposal for a sensorial study to be performed in the following months. Thus, the authors believe adding data from a small-scale study will duplicate our study.
Thank you for understanding.
Dr. Roberta Claro da Silva and authors